# Tips for Managing Ethical Challenges in Advance Care Planning: A Qualitative Analysis of Japanese Practical Textbooks for Clinicians

**DOI:** 10.3390/ijerph19084550

**Published:** 2022-04-09

**Authors:** Yoshihisa Hirakawa, Kaoruko Aita, Mitsunori Nishikawa, Hidenori Arai, Hisayuki Miura

**Affiliations:** 1Department of Public Health and Health Systems, Graduate School of Medicine, Nagoya University, Nagoya 466-8550, Japan; 2Graduate School of Humanities and Sociology, University of Tokyo, Tokyo 113-0033, Japan; aita@l.u-tokyo.ac.jp; 3National Center for Geriatrics and Gerontology, Obu 474-8511, Japan; m-nishik@ncgg.go.jp (M.N.); harai@ncgg.go.jp (H.A.); hmiura@ncgg.go.jp (H.M.)

**Keywords:** advance care planning, dignity, autonomy, end of life, person-centered care, interprofessional ethics

## Abstract

(1) Background: While advance care planning (ACP) provides healthcare professionals with valuable tools to meet patients’ needs in a person-centered manner, several potential ethical challenges are inherent to the process. However, recent studies have largely focused on ACP practicalities such as implementation, execution, and completion rather than on the ethical challenges that clinicians routinely encounter in ACP practices. (2) Research question/aim/objectives: This study aimed to identify tips for clinicians managing ethical challenges in ACP practices. (3) Methods: It performed a brief search for all Japanese published books pertaining to ACP practice available as of January 2021 using the keywords “advance care planning (ACP)” and “autonomy” and analyze the content of nine practical ACP textbooks for clinicians. (4) Results: Two major themes capturing the essential recommendations for managing ethical challenges in ACP were ultimately identified, namely interprofessional ethics and informed consent. (5) Conclusion: The findings suggested tips for managing ethical challenges in ACP: refer to ethical frameworks for interprofessional collaboration and ethical decision making, assess decision-making capacity of family substitute decision makers and one’s eligibility for the role, understand the standard process of informed consent and how to handle situations when the patient are not well informed about the diagnosis and prognosis of non-cancer illness.

## 1. Introduction

Advance care planning (ACP) is a continual process that supports patients, family members, and healthcare professionals (i.e., physicians, nurses, and allied health professionals) in understanding and sharing patients’ personal values, life goals, and preferences regarding future decisions on medical treatment, palliative care, and place of care [1]. ACP generally includes the following specific elements: identifying those individuals who are most likely to benefit from ACP; assessing the person’s capacity for decision making, current condition, and likely prognosis; establishing the person’s health and personal goals, values, and preferences; discussing current and future treatment and personal care options; designating surrogate decision makers in case the patient is no longer able to make decisions; documenting treatment and care plans; and ensuring that these plans are appropriately communicated and available when needed [1,2]. This process is repetitive and typically occurs longitudinally within the context of an existing and continuing patient–family-healthcare professional relationship [1,2,3].

While ACP provides healthcare professionals with valuable tools to meet patients’ needs in a person-centered manner, several potential ethical challenges are inherent in the process [4,5]. Ethical dilemmas and conflicts (i.e., withholding aggressive management, withdrawing life-sustaining treatment, ascertaining patient autonomy) can arise in end-of-life care when a difficult problem cannot be solved in a way that will satisfy everyone involved [5]. Thus, ethical reasoning is a significant component of ACP that affects the actions of patients, their families, and healthcare professionals.

Ethical frameworks are useful perspectives for reasoning on the course of action that provides the most ethical and moral decision making. Several ethical frameworks have been proposed to support decision making in clinical and community settings [6]; however, these frameworks struggle to apply moral theories to concrete ethical problems in real-world clinical settings. While idealized decision theory assumes that all the ethical problems of each case will be considered by healthcare professionals, in reality, some problems are highly likely to be missed or forgotten in busy clinical situations. Clinicians are also not perfectly rational: they may find it difficult to weigh all the relevant information available to them [7] and may need more practical tips on how to reason on ethical problems in ACP.

Despite widespread efforts to promote ACP, the actual completion rate is still low, partly owing to clinicians’ lack of knowledge and awareness of ACP [8,9]. Also due to the absence of legislation specifically encouraging ACP in Japan, it is less practiced in this country than in the United States and the United Kingdom [8], where specific laws and policies on the issue have been adopted. In the United States, for example, the Patient Self-Determination Act was established in 1990 to promote ACP. It requires the certified health care institutions to ask their clients whether they have an advance directive, where patients document their preferences for medical care and appoint a surrogate decision-maker, and inform them of their right to obtain it at the time of enrollment [8]. The UK was an early adopter of ACP nationally, and through the government policy and guidance ACP has been widely recommended, developed and adopted. The two countries gradually have shifted from a legal transactional mode of ACP toward a communications model with a focus on authentic and reliable communication that accurately translates patients’ wishes into the care they receive [10,11]. In contrast, recent Japanese studies have largely focused on ACP practicalities such as implementation, execution, and completion rather than on the ethical challenges that clinicians routinely encounter in ACP practices. The aim of this study is to identify tips for clinicians to manage ethical challenges in ACP practices through an analysis of the content of cases regarding ACP.

## 2. Materials and Methods

### 2.1. Research Design

An analysis of the contents of case studies written in Japanese practical ACP textbooks for clinicians was conducted. A case study is an intensive, systematic investigation of a single individual or group, typically seen in social and life sciences, which is useful for obtaining an in-depth appreciation of a complex and broad issue, event, or phenomenon of interest in its natural, real-life context [12,13]. The analysis had two objectives: (1) to discover the ethical points that need to be addressed in the process of ACP practice and (2) to identify the key aspects of raising ethical awareness in ACP practice among patients, families, and healthcare professionals. Practical textbooks for clinicians were used in the analysis because such textbooks were thought to feature multiple ethics cases along with an explanatory commentary and be written in more didactic manner than academic papers.

#### Participants and Research Context

A brief search for all published Japanese books pertaining to ACP practice available as of the beginning of the study: January 2021, regardless of year of publication, was performed on Amazon Books and Rakuten Books using the keywords “advance care planning (ACP)” and “autonomy”. The titles and customer reviews of the retrieved books were screened for eligibility. Books demonstrating ACP cases in clinical practice were included, while those without cases were excluded. The search was not limited to textbooks but was open to any type of publication, such as handbooks and manuals published in the Japanese language. The search yielded 15 books, of which 9 were relevant for this analysis. The details of these nine books are listed in Table 1.

### 2.2. Analysis

Following the qualitative synthesis of the whole text data of the case presentations (from the description of the patient’s relevant demographic details, medical history, symptoms and signs, treatment or intervention to outcomes, and any other significant details) and the attached expert commentaries, a qualitative content analysis [14] of the qualitative data was undertaken to identify key themes relevant to the topic of the present paper. Because there is no existing theory or framework to start with this analysis, the authors consider inductive approach. The first author (YH), a geriatrician with ample experience in qualitative analysis, conducted line-by-line coding through which pieces of data were segmented and condensed into individual sentences (meaning units). Two nurses and three allied healthcare professionals with experience in ACP practice in the community discussed the emergent meaning units with the first author to perform the final coding check. The grouping process involved reading and comparing individual meaning units, clustering similar units into categories, and inductively formulating themes. An iterative process was used throughout the analysis, and categorical and thematic processes were continued until all six analyzers reached consensus.

### 2.3. Ethical Considerations

This article does not include any studies involving human or animal subjects.

## 3. Results

Two major themes capturing the essential recommendations for managing ethical challenges in ACP were ultimately identified: interprofessional ethics and informed consent.

### 3.1. Interprofessional Ethics

The analysis identified some tips about interprofesional ethics: disclosure of personal information to other individuals or third parties, information exchange among hospitals and primary care settings in ACP, ethical approach to complex decision-making cases, hierarchy breaking for interdisciplinary team meetings, pursuance of team satisfaction.

First, the result of the analysis suggested that healthcare professionals refer to practical guidance on how to share personal information under the Personal Information Protection Law. Sharing personal information with family members and healthcare teams can help optimize end-of-life care by providing a starting point to initiate ACP and relieve patients from having to repeat their stories at each point in the care continuum. Community healthcare professionals play a critical role in sharing useful personal information for ACP among family members and healthcare teams and connecting families and communities in the health system. However, some textbooks in this study pointed out that community healthcare professionals generally shied away from sharing their patients’ personal information with other team members from the perspective of personal information protection. 

“*The personal information protection law likely hinders the support that community professionals could offer individuals living alone*.”(Book No. 3)

“*When it comes to personal information protection, medical ethics rules prevent healthcare professionals from sharing patient information with family members without the patient’s permission*.”(Book No. 7)

Second, the result suggested that healthcare professionals in inpatient hospital settings and community or care home settings work together to exchange ACP information on the transition of care. Changes in patient condition hinder the transition between hospitals, long-term care facilities, and homes during end-of-life care. One of the major barriers to coordinated and effective care transition is poor information exchange between healthcare professionals, which is often exacerbated in community settings. Because patients who engage in ACP are more likely to receive end-of-life care consistent with their values, some textbooks suggest that ACP records should be reviewed when inter-facility care transitions take place.

“*Sharing patients’ ACP between hospital and home-visit healthcare teams becomes increasingly important as end of life draws nearer*.”(Book No. 9)

“*When patients transfer to a new facility, their ACP should be shared and reconfirmed upon admission*.”(Book No. 7)

Third, the result suggested that healthcare professionals use ethical approach to complex decision-making cases. The four pillars of medical ethics—autonomy, beneficence, non-maleficence, and justice—provide a clinical framework for decision making [15]. Because patients’ decision-making capacity and attitudes may fluctuate, some textbooks underscore the importance of reviews of ethical principles and guidelines for good ACP when all efforts to support patients in making their own decisions have failed.

“*The patient’s best interest was discussed among the healthcare professionals, and key points for discussion via the four pillars of medical ethics were identified*.”(Book No. 1)

“*The ethical issues surrounding the decisions made by the patients were discussed among the nurses, along with the ethical principles of nursing and the nursing code of ethics*.”(Book No. 9)

Fourth, the result suggested that healthcare professionals improve interprofessional team meetings by decreasing perceived hierarchy. An ethically problematic clinical case is often used to illustrate the importance of understanding clinical ethics within an interdisciplinary context. A complex case involving the delivery of end-of-life care was used as an example, highlighting how healthcare professionals must work as multidisciplinary and interdisciplinary teams at different times and in parallel with one another. However, some textbooks imply that healthcare settings hold a socio-hierarchical culture with a wide power distance, which makes an interprofessional approach to ACP challenging.

“*The hierarchical power relations between care managers and day-service care workers likely deter the latter from voicing out their opinions*.”(Book No. 3)

“*The healthcare professionals’ paternalistic relationships with their patients and families can affect the patients’ capacity for autonomy in decision-making under a critical situation*.”(Book No. 7)

Finally, the results suggested that healthcare professionals pursue team satisfaction, as well as patient satisfaction. Family caregivers’ satisfaction and healthcare professionals’ job satisfaction are regarded as key aspects of quality of end-of-life care. Although ACP discussions should focus on eliciting patients’ goals and values, some textbooks advocate that decisions need to be satisfactory to all healthcare professionals concerned.

“*Team members should reach satisfactory consensus on ACP in order to avoid ill feelings or legal conflicts among them after the patient’s death*.”(Book No. 4)

### 3.2. Informed Consent

The analysis also identified some tips about informed consent: assessing prognosis by non-physician clinicians, healthcare decision-making capacity, and eligibility to act as informal substitute decision maker when there is no legislation pertaining to proxy decision makers in Japan; delivering bad news to patients about life-threatening chronic illness or conditions other than cancer; having frequent talks with patients and family members about death and dying that trigger many emotional ups and downs.

First, the result of the analysis suggested that healthcare professionals use prognostic indices developed for community-dwelling older individuals. Because patient identification is an important step in ACP discussions [16], the assessment of prognosis is relevant in non-cancer chronic illnesses such as chronic obstructive pulmonary disease as well as in palliative care. Physicians often find it difficult to establish prognoses for non-cancer chronic illnesses compared to cancer. In addition, several textbooks indicate that many non-cancer patients spent the last days of their lives at long-term care facilities, where non-physician stakeholders consistently had difficulty communicating with physicians; thus, a prognosis index such as the “surprise” question “would I be surprised if this patient died within the next year?” helped non-physician stakeholders identify patients with a poor prognosis.

“*Healthcare professionals should be warned not to miss the opportunity to initiate ACP discussions by using a prognosis index such as the surprise question*.”(Book No. 5)

“*Because staging and prognosis assessment are difficult clinical steps in the management of patients living in a nursing home, family members are likely ill-prepared to transfer their loved one to their preferred place of death*.”(Book No. 7)

Second, the result suggested that healthcare professionals consciously assess patient healthcare decision-making capacity and consider eligibility for informal substitute decision makers. Healthcare decision-making capacity is a patient’s ability to understand the benefits and risks of—and alternatives to—a proposed treatment or intervention and to make a choice that is congruent with their own values and preferences. Because it is usually readily apparent for most mentally healthy patients, capacity is assessed intuitively and unconsciously at every medical encounter. However, a more formal and prudent capacity evaluation is essential when there is reason to question a patient’s decision-making abilities, such as when the patient exhibits an acute change in mental status, cognitive impairment, or refusal of a clearly beneficial recommended treatment. If a patient is found not to have the capacity to make decisions, treatment decisions often fall to the closest family member by next-of-kin determination; that is, an informal substitute decision maker should be identified and consulted while no legislation pertaining to proxy decision makers is in place in Japan. In such cases, the substitute decision makers’ capacity also needs to be considered before ACP discussions.

“*When it comes to delivering bad news to family members, healthcare professionals should consider how they have been coping and been supported, and understand that family members have the right not to know this news*.”(Book No. 7)

“*In case family members have been estranged from the patient for many years, healthcare professionals should confirm whether or not they want to act as substitute decision makers*.”(Book No. 3)

Third, the result suggested that healthcare professionals be well equipped to deliver bad news to patients about life-threatening chronic illness or conditions other than cancer. Patients and their families experience differing levels of comfort when discussing end-of-life issues with their healthcare professionals, making this an area that could benefit from interventions targeted at improving the initiation of patient-healthcare professional communication. Some textbooks argue that delivering bad news does not only happen in the context of cancer, further indicating that patients with life-threatening chronic illnesses or conditions other than cancer and their families suffered emotionally from the improper delivery of bad news by healthcare professionals.

“*Upon admission to a nursing home, older residents and their families were routinely asked about ACP; most people complied, but one family refused and voiced strong objections to discussing ACP*.”(Book No. 7)

Finally, the result suggested that health care professionals have frequent talks with patients and family members about death and dying that trigger many emotional ups and downs. Grieving, or the acceptance of death and dying, often progresses through five emotional stages: denial, anger, bargaining, depression, and acceptance. Patients exhibit strong fluctuations in feelings of denial and acceptance of death and dying during the end-of-life period. Some textbooks advocate frequent talks to patients and family members that entail numerous emotional ups and downs throughout the entire process of death and dying. They may otherwise feel they are being unsupportive of their dying loved one with a probability of recovery and survival.

“*A patient who underwent mechanical ventilation gave a Do-Not-Intubate order; however, when his respiratory condition deteriorated again, he changed his mind and asked to be intubated*.”(Book No. 9)

“*Two weeks of life extension using mechanical ventilation may provide sufficient time for family members to prepare for and accept the death of their loved one*.”(Book No. 9)

## 4. Discussion

The findings of the present study suggest that healthcare professionals engaged in ACP should value the importance of interprofessional ethics: ethical frameworks, policies, and procedures of professional practice for multidisciplinary teams in health. Healthcare professionals in the community resort to the services of a variety of professionals and facilities to provide end-of-life care. There is an increasing need for multidisciplinary clinicians skilled in ethical decision making, especially as the interprofessional practice context in the community becomes more complex and concerned with ethical risk management than in hospital settings. The findings also underscored the importance of informal family decision makers’ ability to make medical decisions in tandem with that of the patient because some people require assistance to make decisions due to cognitive impairment or anxiety. Delivering bad news proved to be one of the hardest challenges doctors faced in their practice for end-of-life care of chronic non-cancer illnesses as well as cancer. In end-of-life care settings, healthcare professionals face particular difficulty when diagnosing or predicting the prognosis of a life-threatening, non-cancer illness because of the lack of specialist physicians and the uncertainty in estimating the prognosis of such diseases.

Even though interprofessional collaboration in multidisciplinary teams requires effective information exchange, the study findings suggest that personal information sharing among patients who begin considering ACP and their stakeholders is underpinned by interdisciplinary ethical frameworks. When the need arises for family caregivers or older patients and their healthcare professionals to take a more active role in care, it can be challenging for them to manage their patients’ health information and decision making while also respecting their preferences, privacy, and priorities [17]. Furthermore, the growing number of older people living apart from their informal family decision makers, with whom they often share control of their personal health information and decision making, likely further hinders this process [18].

These findings suggest that ACP information exchange—a key element in maintaining continuity of care among healthcare professionals—often fails during care transitions between settings. These are a well-known cause of health information exchange errors, especially in long-term care settings, partly because older patients are not only more likely to be transferred between multiple facilities but are also more likely to have comorbid conditions, take multiple medications and see several care teams of multidisciplinary professionals [19,20]. This study underscored the importance of ACP information sharing throughout transitions in long-term and end-of-life care and suggested the need for an inter-facility information-sharing system. Regardless of ACP, patients transitioning between care settings experience substantial disruptions in continuity that affect the quality and safety of their care [20]. Several previous studies have advocated the introduction of electronic health records into ACP to share complete, timely, and usable information among hospitals and long-term care facilities to support care transitions [21,22] because information sharing between hospitals and communities is still challenging and often results in poor transitional care processes and outcomes. In addition, this study highlighted a certain hesitation to initiate communication with a perceived higher hierarchy, namely with interprofessional power hierarchies within care teams. Such social gaps create strong nonverbal politeness intended to avoid conflict and maintain harmony [23,24]. In patient care situations, conflicts often occur over ethical principles, especially between beneficence and autonomy. The study findings suggested that healthcare professionals engaging in ACP referred to a series of tools for considering ethics, such as the four basic principles of healthcare ethics [15], when evaluating the merits and difficulties of medical procedures. Medical decision making toward the end of life can involve complex ethical considerations and situations without one “right” way forward [5,25,26]; therefore, complicated moral/ethical issues arise with respect to certain areas of ACP that affect the decisions of patients, families, and healthcare teams. For example, the findings identified the right to refuse the recommended treatment as an ethical consideration for ACP. Although competent patients have the right to refuse treatment, which is a right supported by the ethical principle of autonomy, difficult ethical problems arise when a mentally ill individual needs treatment and refuses it. As another example, this study suggested that such ethical decision-making tools were useful for ACP to help healthcare professionals as well as patients and families work through ethical complexities and ensure that both legal and ethical conflicts are addressed proficiently and to everyone’s satisfaction. Although previous studies have dealt with job satisfaction among healthcare professionals providing end-of-life care [27,28,29,30], to the best of the authors’ knowledge, few ethical tools highlight the satisfaction of end-of-life care teams with respect to ACP discussions.

The results identified several ethical issues related to informed consent. The process requires the patient to be well informed about the diagnosis, nature, purpose, risks, and benefits of the proposed treatments or procedures, alternative treatments or procedures, and risks and benefits of not receiving or undergoing such treatments or procedures [31]. Although accurate diagnosis and prognostification are critical for older people to avoid unnecessary investigations and treatments and the associated costs and harm, the results showed that patients living in the community could not receive relevant, accurate, and unbiased information prior to providing consent due to the uncertainty of non-cancer illness trajectory and the lack of physicians.

In addition to highlighting the importance of the decision-making capacity of family substitute decision makers, the study also underscored the importance of assessing one’s eligibility for this role. There are several reasons that eligibility is essential. First, patients’ spousal caregivers may have suffered cognitive impairment due to advanced age, which could have affected their ability to make decisions on their behalf. Second, circumstances often demand that decisions be made with great haste. Third, in most cases, informal family decision makers may be uncertain about the nuances of their loved one’s wishes because talking about death is still taboo in some families [32] and also because they lack the time and opportunities to discuss death with their loved one living apart from them [18]. Lastly, as emphasized in this study, patients and healthcare professionals often wish to consider family members’ opinions when making medical decisions, and these may possibly take precedence [33,34].

### Study Limitations

This study has several limitations. First, it was a narrative rather than a systematic review, and only two non-academic search engines were used; consequently, other relevant textbooks may have been overlooked. Second, even though acknowledged specialists and authorities made comments on individual textbook cases, the qualitative data may not have been saturated because of space limitations. Third, because the qualitative data were extracted from published textbooks, there may have been some social desirability bias as well as the tendency to underreport socially undesirable comments and to overreport more desirable ones on the individual cases. Finally, because the sample was limited to Japanese textbooks, the findings of this study need to be compared with ones written in English and other languages.

## 5. Conclusions

This qualitative analysis identified two main ethical themes for clinicians managing ethical challenges in ACP practice: internprofessional ethics and informed consent. The findings suggested that healthcare professionals engaged in ACP should value the importance of interprofessional ethics, namely ethical frameworks, policies, and procedures of professional practice for multidisciplinary teams in health. The findings also underscored the importance of informal family decision makers’ ability to make medical decisions in tandem with that of the patient. Patients living in the community could not receive relevant, accurate, and unbiased information prior to providing consent due to the uncertainty of non-cancer illness trajectory and the lack of physicians. Thus, the findings suggested tips for managing ethical challenges in ACP: refer to ethical frameworks for interprofessional collaboration and ethical decision making, assess decision-making capacity of family substitute decision makers and one’s eligibility for the role, understand the standard process of informed consent and how to handle situations when the patient are not well informed about the diagnosis and prognosis of non-cancer illness. The outcomes of this study will contribute to the development of community-based ACP training program that help healthcare professionals to focus on interprofessional ethics and improve the informed consent process.

## Figures and Tables

**Table 1 ijerph-19-04550-t001:** Details of books analyzed in this study.

Book No.	Author/Editor	Freely Translated and Condensed Title	Number of Cases (Relevant Pages)	Place of Publication	Publisher	Year of Publication	ISBN/ASIN	Total Number of Pages
1	Japanese Nursing Association	Person-centered team-based advance care planning	3 (11)	Tokyo	Japanese Nursing Association Publishing	2019	B07RVHGM4Q	119
2	Kazuhiro Nagao	Advance care planning: Getting started guide for home care nurses and care managers	5 (34)	Tokyo	Kenko-to-yoitomodachi	2020	978-4902475098	168
3	Kyoko Oshiro, et al.	Community-based advance care planning: Learning self-determination support from nursing case studies	18 (72)	Tokyo	Nanzando	2020	978-4525500610	146
4	Masako Minooka	Clinical ethics in end-of-life care	6 (43)	Aichi	Nissoken	2020	978-4-7760-1901-5	160
5	Mutsuhito Ui	An overview of advance care planning	4 (10)	Tokyo	Nanzando	2020	978-4525210311	136
6	Masumi Sumita	Empathetic advance care planning: A practical guide for community-based networks of health care, medical and welfare professionals	15 (67)	Tokyo	Medical Friend	2019	978-4839216436	300
7	Mitsunori Nishikawa, et al.	Self-determination support: Learning advance care planning from case studies	24 (93)	Tokyo	Nanzando	2016	978-4525500214	229
8	-	Advance care planning: Process and assessment	3 (30)	Tokyo	Kango no Kagakusha	2020	B08DSYPFWH	99
9	-	Nurse’s role in advance care planning	5 (16)	Tokyo	Kango no Kagakusha	2020	B083JW26JC	99

## Data Availability

Not applicable.

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
