# Peer review of "Tips for Managing Ethical Challenges in Advance Care Planning: A Qualitative Analysis of Japanese Practical Textbooks for Clinicians"

_ijerph, 2022, doi:10.3390/ijerph19084550_

Round 1

Reviewer 1 Report

  Interesting approach on the identification on managing ethical challenges in advance care planning. However, this study deserves some considerations for its publication to be more rigorous:

1) In summary, I believe that the conclusion is not properly explained. The conclusion in the abstract should be the text that is written in lines 317 to 332. 
2) Regarding the purpose of the research in the abstract and line 66, I think the authors intend to identify tips and not extract tips. i think the verb is more appropriate to the study in question;
3) I believe that in the introduction the authors should justify why they opted for a book-based literature review and not a scoping review, based on databases and journal articles;
4) the authors have not specified a time frame for the publication of the books, ten years, twenty years, or any other time frame. They also did not place the date/month when they carried out the research;
5) They also did not put a flowchart of the research. With the eligibility and exclusion criteria they only found nine books or more and they were then discarded?
6) To be rigorous, the analysis is not of the books but of the clinical cases found in the books, so the analysis is on the chapters of the books that have clinical cases on ACP. This should be written down and put on the pages of the books that have been subjected to content analysis;
7) The theoretical framework which sustains the analysis of the texts must be mentioned; themes and sub-themes will emerge but the authors based themselves on some author to proceed to this text analysis. The procedure is explained in line 92 up to line 105 but the theoretical framework of the analysis of the content of the chapters of the book is not mentioned;
8) in line 173 the acronym COPD appears for the first time and should be presented in full;
9) I think that presenting a figure with the themes and sub-themes that emerged, before moving on to the explanation of the results, would help the reader to understand the results;
10) In the limitations of the study, I think it is clear that a study of this nature cannot lead to generalisations. Therefore, the authors should only mention that this work presents some clues for future research to deepen the data. 
11) As for the conclusions, I think that there is more to be developed. What contributions do the authors of this work see for research, for teaching and for society in general?

Author Response

Reviewer 1:

Interesting approach on the identification on managing ethical challenges in advance care planning. However, this study deserves some considerations for its publication to be more rigorous:

1) In summary, I believe that the conclusion is not properly explained. The conclusion in the abstract should be the text that is written in lines 317 to 332.

Author Response) After major revision of conclusion part, I extracted the text from the part as conclusion of the abstract. 

2) Regarding the purpose of the research in the abstract and line 66, I think the authors intend to identify tips and not extract tips. I think the verb is more appropriate to the study in question;

Author Response) As for the objective of this study, Reviewer 3 also recommended to modify the objective of this study, saying “to analyze the content of cases regarding ACP…” is better. I would like to merge the two reviewers’ suggestion and end up with “The aim of this study is to identify tips for clinicians to manage ethical challenges in ACP practices through an analysis of the content of cases regarding ACP”.

3) I believe that in the introduction the authors should justify why they opted for a book-based literature review and not a scoping review, based on databases and journal articles;

Author Response) I described the reason why I chose textbooks instead of academic papers in 2.1 Research design: Practical textbooks were used in the analysis because such textbooks were thought to feature multiple ethics cases along with an explanatory commentary and be written in more didactic manner than academic papers. I deleted the sentence written in 2.1 Research design “Each textbook featured several ethics cases along with an explanatory commentary.”. Even though I was suggested to describe it in the introduction, I feel like it may fit in the research design section.

4) the authors have not specified a time frame for the publication of the books, ten years, twenty years, or any other time frame. They also did not place the date/month when they carried out the research;

Author Response) In Participants and research context section (Method), I have already described the starting month of the study (January, 2021) and the publication year of the relevant books (regardless of year of publication). To be clear, I modify the part like “as of the beginning of the study: January 2021”. I searched the textbook regardless of year of publication, I end up with the past five years shown in Table1.

5) They also did not put a flowchart of the research. With the eligibility and exclusion criteria they only found nine books or more and they were then discarded?

Author Response) I added the explanation for it just before Table 1: The search yielded 15 books, of which 9 were relevant for this analysis. (Six books do not have any cases)

6) To be rigorous, the analysis is not of the books but of the clinical cases found in the books, so the analysis is on the chapters of the books that have clinical cases on ACP. This should be written down and put on the pages of the books that have been subjected to content analysis;

Author Response) To emphasize the point, I modified the aim of this study: “The aim of this study is to identify tips for clinicians to ..... through an analysis of the content of cases regarding ACP.”. Also, “An analysis of the contents of case studies written in Japanese practical ACP textbooks for clinicians was conducted.” was put in 2.1 Research design. As for the pages of the books, I added the number of pages which included case studies in Table1.

7) The theoretical framework which sustains the analysis of the texts must be mentioned; themes and sub-themes will emerge but the authors based themselves on some author to proceed to this text analysis. The procedure is explained in line 92 up to line 105 but the theoretical framework of the analysis of the content of the chapters of the book is not mentioned;

Author Response) Contrary to direct content analysis where researchers have an existing theory or framework that they want to base their research on, conventional content analysis (used in this study) is considered when there is no existing theory or framework to start with. I added one sentence “Because there is no existing theory or framework to start with this analysis, the authors consider inductive approach.”.

8) in line 173 the acronym COPD appears for the first time and should be presented in full;

Author Response) I spelled out COPD as Chronic Obstructive Pulmonary Disease. But this is the only place where COPD is used.

9) I think that presenting a figure with the themes and sub-themes that emerged, before moving on to the explanation of the results, would help the reader to understand the results;

Author Response) Another reviewer (Reviewer3) suggested the same point. He or she also suggested deleting the subthemes per se. Considering both ideas, I decided to delete the subthemes from the text. Instead, in Results section, I added one advice (suggestion) on each paragraph in an organized manner (First, Second, Third,,,,) according to the subthemes (as described above). I hope the revision help the readers to better understand the clinical tips.

10) In the limitations of the study, I think it is clear that a study of this nature cannot lead to generalisations. Therefore, the authors should only mention that this work presents some clues for future research to deepen the data.

Author Response) I notice that the first limitation is similar to the last limitation. Merging your suggestion that generalization is not an appropriate limitation, I end up with “Finally, because the sample was limited to Japanese textbooks, the findings of this study need to be compared with ones written in English and other languages.”.

11) As for the conclusions, I think that there is more to be developed. What contributions do the authors of this work see for research, for teaching and for society in general?

Author Response) After major revision of conclusion section based on the reviewers’ suggestion, as future direction, I added “The outcomes of this study will contribute to the development of community-based ACP training program that help healthcare professionals to focus on interprofessional ethics and improve the informed consent process.”.

Reviewer 2 Report

Thank you for the opportunity to review this manuscript. This is a significant interest to clinicians in Japan only. Perhaps a comparison with USA and UK ACP knowledge may provide Japanese clinicians and provide some future focus, you may wish to consider this in your next paper.

Title: Considering you have engaged with Japanese textbooks only including Japanese Practical Textbooks would be more explicit to the reader.

Key Words: ACP should be included

Line 39 - Family needs to be included and explanation of the relationship between family, patient and clinician with developing ACP

Line 69 and 74 - include content of line 74 before line 69

Line 224 - A cultural consideration within Japan would assist in this section

throughout text quotes are used but not included

Author Response

Reviewer 2:

Thank you for the opportunity to review this manuscript. This is a significant interest to clinicians in Japan only. Perhaps a comparison with USA and UK ACP knowledge may provide Japanese clinicians and provide some future focus, you may wish to consider this in your next paper.

Author Response) Thank you for appreciating the value of the study. Because your idea is interesting to me, I would like to expand the relevant papers to the other countries such as US, UK, and other western countries.

Title: Considering you have engaged with Japanese textbooks only including Japanese Practical Textbooks would be more explicit to the reader.

Author Response) I added “Japanese” onto the title.

Key Words: ACP should be included

Author Response) I added “advance care planning” to Key Words list.

Line 39 - Family needs to be included and explanation of the relationship between family, patient and clinician with developing ACP

Author Response) I completely agree with it. I insert “family” between patient-healthcare professional relationship.

Line 69 and 74 - include content of line 74 before line 69

Author Response) I think that this comment is accordance with Reviewer 3’ s comment “The objective should be at the end of the introduction and not in the Methods section”. I did so according to your suggestion.

Line 224 - A cultural consideration within Japan would assist in this section

Author Response) I added the sentence “They may otherwise feel they are being unsupportive of their dying loved one with a probability of recovery and survival.”. In Japan, some family caregivers feel guilty of leaving their dying loved one alone or abandoning the resuscitation attempt.

throughout text quotes are used but not included

Author Response) As suggested here, I modified the style of presentation: all the quotes are separated from the sentences in a popular writing way of qualitative research results.

Reviewer 3 Report

Many thanks for the opportunity to review this manuscript.

This is an interesting paper that addresses a pertinent issue. I would like to make a few suggestions to improve the quality of the paper before its publication.

Introduction: It requires a more detailed exploration of international literature regarding advanced care planning (APC).

Objective: To extract tips for clinicians to manage ethical challenges in ACP practices.

This is not an appropriate objective for a research study. I recommend: ‘to analyse the content of cases regarding APC…’. The advice could be provided in a section about implications for practice after the conclusions. Furthermore, the Results section presents emerging themes of a thematic analysis of the selected books rather than advice.

The objective should be at the end of the introduction and not in the Methods section.

Design: The Abstract presents the study as a literature review and the Methods section as a literature review of case studies. However, in other sections of the manuscript, it is described as a qualitative study (line 326). I do not think that this is a case study so it should be presented as a qualitative analysis of content (example: https://doi.org/10.1080/14681811.2017.1308858)

Results: There are two themes and 11 subthemes but only the themes are discussed. There should be a table or graph with the themes and subthemes. The results should be structured around these themes and subthemes. An alternative would be to not make reference to the subthemes as they are not seen elsewhere in the document.

I would add a paragraph on implications for practice in the Conclusions section, where one could provide the tips that are promised in the title and objective.

Author Response

Reviewer 3:

Many thanks for the opportunity to review this manuscript.

This is an interesting paper that addresses a pertinent issue. I would like to make a few suggestions to improve the quality of the paper before its publication.

Author Response) Thank you for appreciating the value of the study.

Introduction: It requires a more detailed exploration of international literature regarding advanced care planning (ACP).

Author Response) I should be clear why in such western countries ACP is more practiced. So, I modified the sentence while deleting “Western countries such as” and “Sweden” and adding “where specific laws on the issue have been adopted”. Because I cannot find the literature Sweden has specific legislation, I delete the country from the example.

Objective: To extract tips for clinicians to manage ethical challenges in ACP practices.

This is not an appropriate objective for a research study. I recommend: ‘to analyse the content of cases regarding APC…’. The advice could be provided in a section about implications for practice after the conclusions. Furthermore, the Results section presents emerging themes of a thematic analysis of the selected books rather than advice.

Author Response) As for the objective of this study, Reviewer 1 also recommended to modify the objective of this study, saying “to identify tips” is better. I would like to merge the two reviewers’ suggestion and end up with “The aim of this study is to identify tips for clinicians to manage ethical challenges in ACP practices through an analysis of the content of cases regarding ACP”. As for the conclusion section, the suggestion is also given in the last comment as below. I respond to the suggestion there. In Results section, I added one advice (suggestion) on each paragraph in an organized manner (First, Second, Third,,,,).

The objective should be at the end of the introduction and not in the Methods section.

Author Response) According to the suggestion, I moved it to the end of the introduction section.

Design: The Abstract presents the study as a literature review and the Methods section as a literature review of case studies. However, in other sections of the manuscript, it is described as a qualitative study (line 326). I do not think that this is a case study so it should be presented as a qualitative analysis of content (example: https://doi.org/10.1080/14681811.2017.1308858)

Author Response) Thank you for your comments and valuable information. According to them, I become consistent with the terms used in the text while emphasizing “analysis” instead of “literature review”. I also deleted the following from Method section (Research Design); therefore, it is ideal for practice-oriented research. The literature review was used for the qualitative data collection of ACP cases.

Results: There are two themes and 11 subthemes but only the themes are discussed. There should be a table or graph with the themes and subthemes. The results should be structured around these themes and subthemes. An alternative would be to not make reference to the subthemes as they are not seen elsewhere in the document.

Author Response) As suggested, I delete the subthemes from the text. Instead, in Results section, I added one advice (suggestion) on each paragraph in an organized manner (First, Second, Third,,,,) according to the subthemes (as described above).

I would add a paragraph on implications for practice in the Conclusions section, where one could provide the tips that are promised in the title and objective.

Author Response) Considering the title and aim of this study, I focused on “tips” and revised the conclusion section. I added “Patients living in the community could not receive relevant, accurate, and unbiased information prior to providing consent due to the uncertainty of non-cancer illness trajectory and the lack of physicians.” and “Thus, the findings suggested tips for managing ethical challenges in ACP: refer to ethical frameworks for interprofessional collaboration and ethical decision making, assess decision-making capacity of family substitute decision makers and one’s eligibility for the role, understand the standard process of informed consent and how to handle situations when the patient are not well informed about the diagnosis and prognosis of non-cancer illness.”, while I deleted “Thus, ACP can be challenging for everyone concerned and involves complex ethical considerations and situations in which ethical frameworks are useful for assisting stakeholders in comprehensively discussing ethical cases without omitting important discussion points.”. 

Round 2

Reviewer 3 Report

Thank you for sending me this revised paper.
The article has improved and gained methodological consistency.
Most of the recommendations have been addressed. However, the response to the first recommendation is not consistent with the content of the recommendation. It asked for "a more detailed exploration of international literature regarding advanced care planning". The response has been the modification of a sentence on Sweden?
The request was for the authors to expand the literature on advanced care planning in the world by including some more studies of advanced care planning in other countries.

Author Response

Reviewer3’s comment:

The article has improved and gained methodological consistency.

Most of the recommendations have been addressed. However, the response to the first recommendation is not consistent with the content of the recommendation. It asked for "a more detailed exploration of international literature regarding advanced care planning". The response has been the modification of a sentence on Sweden?

The request was for the authors to expand the literature on advanced care planning in the world by including some more studies of advanced care planning in other countries

Author Response) I am sorry for my misunderstanding of the suggested point. Now I added the background information with additional references [10,11]: In the United States, for example, the Patient Self-Determination Act was established in 1990 to promote ACP. It requires the certified health care institutions to ask their clients whether they have an advance directive, where patients document their preferences for medical care and appoint a surrogate decision-maker, and inform them of their right to obtain it at the time of enrollment [8]. The UK was an early adopter of ACP nationally, and through the government policy and guidance ACP has been widely recommended, developed and adopted. The two countries gradually have shifted from a legal transactional mode of ACP toward a communications model with a focus on authentic and reliable communication that accurately translates patients’ wishes into the care they receive [10,11]. In contrast,.......

  1. Sabatino, CP. The evolution of health care advance planning law and policy. Milbank Q. 2010, 88, 211-239.
  2. Cairns R. Advance care planning: thinking ahead to achieve our patients' goals. Br J Community Nurs. 2011, 16, 427.
